# Examining person-centered maternity care in a peri-urban setting in Embakasi, Nairobi, Kenya

**Jackline Oluoch-Aridi** [1]*, **Patience Afulani**[2], **Cindy Makanga**[1], **Danice Guzman**[1], **Laura Miller-Graff**[1,3]

**1** The Ford Family Program in Human Development Studies and Solidarity, Kellogg Institute for International Studies, University of Notre Dame, Nairobi, Kenya, **2** Department of Epidemiology & Biostatistics and Obstetrics, Gynecology & Reproductive Sciences, University of California, San Francisco (UCSF), San Francisco, California, United States of America, **3** Kroc Institute for International Peace Studies and Department of Psychology, University of Notre Dame, Notre Dame, Indiana, United States of America

* Joluocha@nd.edu

## Abstract

**Data Availability Statement:** The data from this study can be found at the following repository: https://doi.org/10.5061/dryad.s1rn8pk7w.

### Introduction

Peri-urban settings have high maternal mortality and the quality of care received in different types of health facilities is varied. Yet few studies have explored the construct of person-centered maternity care (PCMC) within peri-urban settings. Understanding women's experience of maternity care in peri-urban settings will allow health facility managers and policy makers to improve services in these settings. This study examines factors associated with PCMC in a peri-urban setting in Kenya.

### Methods and materials

We analyzed data from a cross-sectional study with 307 women aged 18–49 years who had delivered a baby within the preceding six weeks. Women were recruited from public (*n* = 118), private (*n* = 76), and faith based (*n* = 113) health facilities. We measured PCMC using the 30-item validated PCMC scale which evaluates women's experiences of dignified and respectful care, supportive care, and communication and autonomy. Factors associated with PCMC were evaluated using multilevel models, with women nested within facilities.

### Results

The average PCMC score was 58.2 (SD = 13.66) out of 90. Controlling for other factors, literate women had, on average, about 6-point higher PCMC scores than women who were not literate (β = 5.758, p = 0.006). Women whose first antenatal care (ANC) visit was in the second (β = -5.030, p = 0.006) and third trimester (β = -7.288, p = 0.003) had lower PCMC scores than those whose first ANC were in the first trimester. Women who were assisted by an unskilled attendant or an auxiliary nurse/midwife at birth had lower PCMC than those assisted by a nurse, midwife or clinical officer (β = -8.962, p = 0.016). Women who were

**Funding:** The authors JOA LMG, DB and CM received funding from the Hellen Kellogg Institute for International studies. The funders had no role in study design, data collection and analysis, decision to publish, or preparation of the manuscript.

**Competing interests:** The authors have declared that no competing interests exist.

interviewed by phone (β = -7.535, p = 0.006) had lower PCMC scores than those interviewed in person.

## Conclusions

Factors associated with PCMC include literacy, ANC timing and duration, and delivery provider. There is a need to improve PCMC in these settings as part of broader quality improvement activities to improve maternal and neonatal health.

## Introduction

An estimated 303,000 maternal deaths occurred globally in 2015, with 66.3% of these deaths occurring in sub-Saharan Africa [1]. Kenya's maternal mortality ratio is estimated at 362 deaths per 100,000 live births [2]. Some of the maternal deaths in Kenya have been attributed to childbirth outside of health facilities. The latest Kenya Demographic Health Survey (KDHS), showed a modest increase in the proportion of women delivering in a health facility from 44% in 2009 to 61% in 2014 [2]. Previous work suggests that poor quality of maternity care, including fear of abuse and disrespect perpetuated by health workers, is a potential reason for women's decision to deliver outside of health facilities in Kenya [3,4].

Person-centered maternity care (PCMC) represents the interpersonal dimensions of quality of care, which is critical to experience of care. It refers to care that is respectful and responsive to the needs of women and their families [5]. PCMC emphasizes the patient-provider relationship, and highlights issues such as whether women are treated in a dignified manner, are communicated to effectively, and feel involved in decision-making about their care. It also includes emotional and social support during childbirth by health care providers [5]. PCMC extends prior frameworks such as that by Bowser and Hill (2010) on disrespect abuse and that by Bohren and colleagues (2014) [11] on mistreatment, to highlight that women's childbirth experiences fall on a continuum from the best to the worst possible experiences [6]. Categories of disrespect and abuse such as: physical abuse, non-consented care, non-confidential care, undignified care, discrimination, abandonment, and detention in health facilities [7] therefore represent poor PCMC. Studies exploring perceptions and experiences of women and health care workers in diverse sub-Saharan African countries such as South Africa, Nigeria, Guinea, and Tanzania provide evidence that poor person-centered maternity care during facility-based childbirth is prevalent and is a growing concern [8–11].

Most studies examining women's experience during delivery in Kenya use qualitative methods, which provide vivid descriptions of the manifestations of disrespect and abuse [4,12]. One study conducted in a small rural public hospital in Western Kenya, describes women's experiences with the health system as unsatisfactory—with women expressing frustration with lack of confidentiality, lack of autonomy, abandonment by providers, and dirty maternity care settings [13]. A few quantitative studies also provide insight into the prevalence of disrespect and abuse in health facilities in Kenya. For instance, one study found that about 20% of women report some kind of undignified care, including non-confidential care, neglect, non-consented care, or physical abuse [14]. A mixed methods study in a rural county in western Kenya highlights poor PCMC across various domains, including poor communication and inadequate support during childbirth [6,15].

Although previous studies have examined PCMC in both rural and urban settings in Kenya [5,16], little work has been done in peri-urban settings. Peri-urban settings in Kenya are often

close to cities, but lack access to basic amenities such as running water and adequate sanitation. The healthcare system within these settings has also been reported to be deficient and health facilities are known to provide varied quality of care [17]. Low income women in such settings often have high levels of mortality despite delivering in a health facility [18,19].

Recent qualitative work in a peri-urban setting in Nairobi documents mistreatment of women during childbirth, as well as perceptions by healthcare workers that the health system is weak and under-supported [20]. Studies in India also highlight how mistreatment of women living in urban slums affects decision-making for facility-based childbirth [21]. The poor PCMC in peri-urban settings places women at greater risk for not accessing health care, and encourages home deliveries that may pose health risks to women and their children [20]. Despite the qualitative evidence, there remains a dearth of quantitative studies that empirically assess the extent to which women receive person-centered care during childbirth in peri-urban settings.

Previous attempts to measure women's experiences during childbirth use binary measures of disrespect and abuse. In order to better assess the quality of maternity care, Afulani et al. (2017) developed the PCMC scale to assess women's experience of labor and delivery services along three dimensions: dignity and respect, communication and autonomy, and supportive care [5]. The 30-item scale was initially validated in rural and urban Kenya [5] and subsequently in India and Ghana [22,23]. Research based on this scale shows that across these settings PCMC is often suboptimal and the most disadvantaged women received the poorest PCMC [24]. The PCMC scale has also been used to assess the factors that affect PCMC as well as consequences of PCMC [16,25].

We intend to extend the literature in two ways. First, the PCMC scale is used to evaluate PCMC in a peri-urban setting, providing important data on PCMC in a new context. We hypothesize, based on previous qualitative work, that PCMC will also be low in this setting. Second, we examine sociodemographic and facility characteristics associated with PCMC, with the goal of better informing how practice and policy might advance maternity care in settings with significant development challenges. The findings provide an important contribution to the growing literature on PCMC, and will help guide quality improvement initiatives to improve women's experience during facility-based childbirth.

## Methods

### Study setting

This study is a cross-sectional study on perceived quality of maternity care in the peri-urban setting of *Embakasi* within Nairobi City in Kenya. Nairobi County is the most populous county in Kenya with a population of close to 4.4 million [26]. *Embakasi* area is the most populous area within Nairobi, with 5 sub-counties and a population of almost one million people [27]. The area is characterized by low-income housing and informal settlements with poor access to water and waste disposal. The largest garbage dumping site for the city of Nairobi is situated in one of the sub-counties of *Embakasi*. The health system within *Embakasi* consists of public hospitals, health centers, and several private and faith-based health facilities.

### Data collection

Study data were collected between January and May 2020. In order to reflect women's experiences across all types of health facilities in the area, women were recruited from three types of health facilities: public, private, and faith-based facilities. The women were recruited using a multistage purposive sampling approach from the sub-County level. First, the *Embakasi* area was divided into its constituent sub-Counties. We then selected health facilities that were representative of the different types of health facilities in each sub-County. With the assistance of

health facility management, women aged between 18 and 49 years, who had delivered within six weeks preceding the study were recruited at postnatal clinics. All women provided written or verbal informed consent to be interviewed. The interviews were conducted by the first author and three research assistants who were trained in research ethics and study procedures in either English or Swahili, depending upon participant preference. Interviews were conducted in private spaces at the respective health facilities, by phone, or in the respondent's community. Variation in location of data collection was due to restrictions in movement due to COVID-19, and other logistical concerns. 320 women were approached for the interviews and 307 agreed to be interviewed representing a response rate of 96%. The women were compensated $10 for the interview to cover transportation costs to the interview venue.

## Ethics approval

Ethics approval for the study was provided by the Strathmore University Institutional Ethics Review Committee (SU-IERC) and the University of Notre Dame Institutional Review Board. The study was also approved by the National Commission for Science and Technology (NACOSTI) and the Director of health services in the sub-county.

## Measures

**Dependent variable: The person-centered maternity score (PCMC score).** The PCMC scale is a validated 30-item scale with three sub-scales for i) dignity and respect, ii) communication and autonomy, and iii) supportive care. Each item is on a 4-point response scale with response options as "no, never" (coded 0), "yes, a few times" (1), "yes, most of the time" (2), and "yes all the time" (3). The full list of items is provided in additional file 1. Prior validation showed the scale has high content, construct, and criterion validity and with good internal consistency reliability [16]. Cronbach's alpha for the 30 items is 0.89. Summing response to the items (after reverse coding negatively worded items) yields a score range of 0 to 90, with lower scores implying poorer PCMC. To account for missing responses to questions which were not applicable to certain women (e.g. women who delivered via elective cesarean section did not have to answer questions on their experience during labor) the scores were calculated using a running mean across items, and then rescaled to reflect a standard range (0 to 90) to enable comparisons to previously published work on the scale [16,24]. All sub-scale scores were standardized to range from 0 to 100 to enable comparisons across sub-scales.

**Independent variables.** *Participant characteristics*. This included sociodemographic factors that might affect the quality of PCMC that a woman receives—such as age, parity, marital status, religion, and tribe. We also assessed socioeconomic factors such as education, literacy, woman and partner's occupation status, wealth quintile, and empowerment. Education was categorized as no school/primary, post primary/vocational/secondary, and college. Literacy was assessed through a survey question asking if the woman reads with difficulty or is illiterate, versus if the woman reads very well. The woman and her spouse's employment status were assessed by a survey question asking, "Do you do any work for which you are paid?" and "Does your spouse/partner do any work for which he is paid?" Household wealth was measured in quintiles and calculated from an urban wealth index based on 13 questions on household assets [28]. Empowerment was assessed using questions from the Demographic Health Survey (DHS) module that measures sociocultural empowerment, including attitudes regarding gender norms and gender-based violence [29]. The scores are divided into low or high empowerment, using the median score. We also included a measure of experience of intimate partner violence which has been found to be associated with PCMC prior studies [16]. Responses indicating exposure to any of the items resulted in a code of "yes" for exposure to IPV.

*Facility and provider characteristics.* The facility where the woman delivered was classified as a government hospital (higher level), health center (lower level), or private/faith-based health facility. Provider type indicates the highest skilled provider who attended at the delivery. Responses were categorized as low or no skill (auxiliary nurse or midwife, friend, relative or no one), skilled (clinical officer, nurse or midwife), or high skilled (doctor). Sex of provider indicates the reported sex of the highest skilled provider (male, female, or refused/delivered alone).

*Other covariates.* To assess potential impact of familiarity and prior contact with the health system, we included assessments of whether women had previously delivered at a health facility and the timing and frequency of antenatal care. We also included a variable on whether the respondent had experienced any complications during her pregnancy and delivery, and if she perceived the complication as severe. Finally, we controlled for the timing and location of the interview.

## Statistical analysis

We first conducted descriptive analysis of all study variables. We then examined bivariate differences in PCMC scores by the independent variables using cross-tabulations and simple Ordinary Least Squares (OLS) regression with robust standard errors, clustered at the level of the health facility. Finally, we conducted multivariate analysis using multilevel models (MLM), with participants nested within health care facilities. MLM improves the specification of between and within facility effects, through the inclusion of random intercepts accounting for between-facility effects and fixed effects for facility type. The model was fitted via restricted maximum likelihood (REML), due to the relatively small number of health facilities. Individual-level sociodemographic characteristics and individual experiences of labor and delivery (e.g., professional status of personnel delivering child) were entered as level-1 predictors, and facility type (private, public, faith-based) was entered as a level-2 predictor. Only variables that were significantly associated with PCMC scores in the bivariate models or in previous studies were included in the MLMs. With this shortened list of variables, we ran tests of collinearity using the variance inflation factor (VIF), and eliminated variables which were highly correlated with other variables in the model. Initial models produced VIFs ranging from 1.17 to 10.95. In the final model, the VIFs ranged from 1.17 to 3.85, indicating a reduction in potential collinearity. The intraclass correlation coefficient in the final MLM was 0.176, suggesting that the nested model is more appropriate for the data.

## Results

### Sociodemographic characteristics

Table 1 shows the univariate and bivariate distributions of the respondent's sociodemographic characteristics. About 74% of the respondents were under 29 years old, and 85% were ever married. The average parity was two, with only 14% of the women having four or more children. About 45% reported post-primary education, and almost half (49%) of their partners had post-primary education. Most (85%) of the women were literate and read very well. About a third (37%) belonged to the highest urban wealth quintile, although the majority (88%) are unemployed. About a quarter of were of the Luo tribe, with other Christian groups (apart from Catholic and Protestants) being the major religion (59%). Most (77%) delivered at a public or faith-based health facility, with only 23% delivering at private health facilities. About half (55%) were classified as having high empowerment based on the empowerment measures used, although 43% had experienced intimate partner violence. Most women reported that their deliveries were attended by a woman (72%) and most (96%) identified the highest skilled attendant present at their delivery as a doctor, clinical officer or nurse/midwife. A majority

**Table 1. Respondent's sociodemographic characteristics and bivariate associations with PCMC.**

| Variable | Descriptive statistics | | | | Bivariate associations with PCMC scores | | | |
| | | | | | Crosstabs | | OLS Bivariate Regressions | |
| | Frequency | % | Mean PCMC score | SD | Coefficient | | Confidence Interval | |
| *Age* | | | | | | | | |
| 18–24 | 118 | 38.4 | 57.6 | 12.8 | [omitted] | | | |
| 25–29 | 110 | 35.8 | 57.7 | 13.7 | 0.0924 | | -3.646 | 3.831 |
| 30 and older | 79 | 25.7 | 57.3 | 12.4 | -0.342 | | -2.734 | 2.050 |
| *Marital Status* | | | | | | | | |
| Never Married | 45 | 14.7 | 56.5 | 13.4 | [omitted] | | | |
| Ever Married | 262 | 85.3 | 57.8 | 13.0 | 1.035 | | -2.670 | 4.741 |
| *Number of births* | | | | | | | | |
| 1 | 83 | 27 | 58.1 | 12.9 | [omitted] | | | |
| 2 | 116 | 37.8 | 58.8 | 12.2 | 0.779 | | -2.308 | 3.867 |
| 3 | 66 | 21.5 | 56.0 | 14.1 | -2.320 | | -9.783 | 5.143 |
| 4 or more | 42 | 13.7 | 55.7 | 13.7 | -2.687 | | -8.241 | 2.867 |
| *Education Level* | | | | | | | | |
| No School/Primary | 121 | 39.4 | 56.7 | 12.8 | [omitted] | | | |
| Post-primary/Vocational/Secondary | 139 | 45.3 | 58.1 | 12.6 | 1.473 | | -1.180 | 4.127 |
| College or university | 47 | 15.3 | 58.4 | 14.7 | 1.815 | | -5.283 | 8.913 |
| *Education Level of Partner* | | | | | | | | |
| No School/Primary | 64 | 20.8 | 58.9 | 10.8 | [omitted] | | | |
| Post-primary/Vocational/Secondary | 149 | 48.5 | 57.7 | 13.5 | -1.231 | | -7.985 | 5.524 |
| College or university | 53 | 17.3 | 55.4 | 13.9 | -3.384 | | -9.928 | 3.160 |
| No partner | 41 | 13.4 | 57.8 | 13.1 | -0.980 | | -9.253 | 7.293 |
| *Literacy* | | | | | | | | |
| Illiterate or reads with difficulty | 45 | 14.7 | 52.9 | 12.8 | [omitted] | | | |
| Yes, very well | 262 | 85.3 | 58.4 | 12.9 | 5.399* | | 0.0615 | 10.74 |
| *Wealth Quintile (Urban)* | | | | | | | | |
| Poor or Poorer | 85 | 27.7 | 56.6 | 12.2 | [omitted] | | | |
| Middle | 107 | 34.9 | 58.9 | 11.7 | 2.660** | | 1.024 | 4.296 |
| Richer or Richest | 115 | 37.5 | 57.0 | 14.6 | 0.446 | | -2.436 | 3.328 |
| *Occupation* | | | | | | | | |
| Not Employed | 269 | 87.6 | 58.0 | 12.8 | [omitted] | | | |
| Employed | 38 | 12.4 | 54.5 | 14.4 | -3.556 | | -8.856 | 1.744 |
| *Partner's Occupation* | | | | | | | | |
| Agriculture or casual labor | 114 | 37.1 | 58.6 | 11.4 | [omitted] | | | |
| Salaried or self-employed | 144 | 46.9 | 56.9 | 14.1 | -2.265 | | -5.645 | 1.114 |
| Unemployed or no partner | 49 | 16 | 57.3 | 13.2 | -1.536 | | -5.978 | 2.906 |
| *Works for Health Facility* | | | | | | | | |
| No | 279 | 90.9 | 57.9 | 12.7 | [omitted] | | | |
| Yes | 28 | 9.1 | 53.9 | 15.6 | -4.294* | | -8.533 | -0.0538 |
| *Experienced any domestic violence* | | | | | | | | |
| No | 176 | 57.3 | 57.7 | 13.6 | [omitted] | | | |
| Yes | 131 | 42.7 | 57.4 | 12.2 | -0.500 | | -2.907 | 1.908 |
| *Empowerment* | | | | | | | | |
| Low empowerment | 138 | 45 | 56.9 | 13.1 | [omitted] | | | |
| High empowerment | 169 | 55 | 58.1 | 12.9 | 0.960 | | -2.746 | 4.666 |

*(Continued)*

**Table 1.** (Continued)

| Variable | Descriptive statistics | | | Bivariate associations with PCMC scores | | | |
|---|---|---|---|---|---|---|---|
| | | | | Crosstabs | | OLS Bivariate Regressions | |
| | Frequency | % | Mean PCMC score | SD | Coefficient | Confidence Interval | |
| *Highest skilled person at delivery* | | | | | | | |
| Auxiliary Nurse, Auxiliary Midwife or No Skilled person | 12 | 3.9 | 43.9 | 18.7 | [omitted] | | |
| Clinical Officer, Nurse, Midwife | 171 | 55.7 | 57.9 | 11.4 | 13.75* | 2.816 | 24.68 |
| Doctor | 124 | 40.4 | 58.5 | 13.8 | 14.85* | 3.791 | 25.92 |
| *Gender of Main person who assisted delivery* | | | | | | | |
| Man | 84 | 27.4 | 56.2 | 14.2 | [omitted] | | |
| Woman | 220 | 71.7 | 58.6 | 11.8 | 2.442 | -1.383 | 6.267 |
| Refused or Delivered alone | 3 | 1 | 20.3 | 9.0 | -36.48*** | -45.82 | -27.15 |
| *Pregnancy Complications* | | | | | | | |
| No | 228 | 74.3 | 58.9 | 11.6 | [omitted] | | |
| Yes | 79 | 25.7 | 53.7 | 15.7 | -6.001 | -13.57 | 1.565 |
| *Severe Pregnancy Complications* | | | | | | | |
| No | 263 | 85.7 | 58.1 | 12.5 | [omitted] | | |
| Yes | 44 | 14.3 | 54.3 | 15.5 | -4.356 | -9.781 | 1.070 |
| *Previously Delivered in a Health Facility* | | | | | | | |
| No | 109 | 35.5 | 57.2 | 12.9 | [omitted] | | |
| Yes | 198 | 64.5 | 57.8 | 13.1 | 0.338 | -1.895 | 2.572 |
| *Trimester of first Antenatal visit* | | | | | | | |
| First | 62 | 20.2 | 62.8 | 10.4 | [omitted] | | |
| Second | 190 | 61.9 | 56.8 | 12.6 | -6.302 | -13.48 | 0.879 |
| Third | 55 | 17.9 | 54.3 | 15.3 | -9.030 | -22.15 | 4.093 |
| *Number Antenatal Visits* | | | | | | | |
| Less than 4 (or don't remember) | 110 | 35.8 | 55.7 | 14.5 | [omitted] | | |
| 4 or more | 197 | 64.2 | 58.6 | 12.0 | 3.516 | -1.915 | 8.947 |
| *Post-partum length* | | | | | | | |
| Less than 5 weeks | 150 | 48.9 | 60.2 | 11.5 | [omitted] | | |
| 5 weeks or more | 157 | 51.1 | 55.1 | 13.9 | -5.446** | -8.061 | -2.831 |
| *Religion* | | | | | | | |
| Catholic | 72 | 23.5 | 57.1 | 13.4 | [omitted] | | |
| Protestant/Pentecostal | 48 | 15.6 | 58.6 | 13.3 | 1.749 | -3.411 | 6.908 |
| Other Christian | 179 | 58.3 | 57.0 | 12.8 | -0.580 | -3.016 | 1.855 |
| Muslim, other religion or refused | 8 | 2.6 | 67.8 | 7.6 | 10.56 | -3.678 | 24.80 |
| *Tribe* | | | | | | | |
| Luo | 78 | 25.4 | 55.2 | 13.0 | [omitted] | | |
| Kikuyu | 62 | 20.2 | 56.9 | 14.7 | 1.628 | -2.640 | 5.897 |
| Luhya | 67 | 21.8 | 59.1 | 11.9 | 3.957* | 0.419 | 7.496 |
| Kamba | 54 | 17.6 | 59.3 | 13.7 | 4.669* | 0.783 | 8.554 |
| Other or refused | 46 | 15 | 58.3 | 11.0 | 3.598* | 0.0708 | 7.125 |
| *Location of Interview* | | | | | | | |
| Health facility | 25 | 8.1 | 61.4 | 14.5 | [omitted] | | |
| In the community/a home | 79 | 25.7 | 52.6 | 14.7 | -10.13*** | -14.08 | -6.172 |
| Phone | 203 | 66.1 | 59.0 | 11.6 | -3.317* | -5.987 | -0.647 |
| *Type of Facility* | | | | | | | |
| Public | 116 | 37.8 | 51.1 | 14.3 | [omitted] | | |

*(Continued)*

**Table 1.** (Continued)

| Variable | Descriptive statistics | | | Bivariate associations with PCMC scores | | | |
| --- | --- | --- | --- | --- | --- | --- | --- |
| | | | | Crosstabs | | OLS Bivariate Regressions | |
| | Frequency | % | Mean PCMC score | SD | Coefficient | Confidence Interval | |
| Faith-based | 119 | 38.8 | 62.7 | 9.6 | 12.15*** | 12.15 | 12.15 |
| Private | 72 | 23.5 | 59.5 | 11.2 | 8.373* | 0.550 | 16.20 |

SD = Standard Deviation;

*** p<0.001,

** p<0.01,

* p<0.05.

(74%) reported no complications during their pregnancy and delivery. Sixty-five percent reported having previously delivered at a health facility. Most of the interviews were conducted via phone (66%).

**PCMC.** The individual items in the PCMC scale and sub-scale are shown in Table 2. The average PCMC score was 58.2 out of 90 (SD = 13.7; Range = 11–85). The average sub-scale score was 14.7 (SD = 3.17; Range 2–18) for Dignity and Respect, 15.74 (SD = 4.9; range 2–27) for Communication and Autonomy, and 27.76 (SD = 7.2; range 4–45) for Supportive Care. The standardized scores are shown on Table 3. Some notable findings from the individual items regarding PCMC in this context include the presence, albeit low prevalence, of physical (5%) and verbal (10%) abuse. Further, the majority of respondents (74%) in this study reported that health care workers never introduced themselves and about one fifth (22%) reported that the health care workers did not call them by name. The presence of supportive care was also sub-optimal. In particular, a large proportion of respondents were not allowed to have a companion during labor (78%) and delivery (84%).

### Bivariate results

The bivariate results are shown in Table 1. Without accounting for other factors, women who had their first antenatal visit in the first trimester had higher mean PCMC scores than women who started ANC in later trimesters. Also, women who read very well had higher PCMC scores than those who were illiterate or read with difficulty. Women in the middle wealth quintile had higher scores than those in the poor or poorer quintiles. Women whose births were attended by a skilled professional scored higher on the PCMC scale than those few whose births were attended by a low skilled or unskilled person. PCMC scores were also lower with higher postpartum length. Tribe demonstrated some correlation with PCMC score in the bivariate analysis, with women identifying as Luhya and Kamba scoring higher than women identifying as Luo. Finally, location of interview was correlated with PCMC score, with women interviewed in the community or home reporting lower scores than those interviewed in the facility.

### Multilevel model

The null multilevel model had an intraclass correlation coefficient (ICC) of 0.170 (95% CI 0.053–0.430), suggesting that there was significant variation in women's reports of PCMC across facilities, and that nesting is required. The intraclass correlation coefficient in the final model was 0.144 [95% CI .024, .531]. After accounting for other factors (See Table 4), women

**Table 2. Distribution of the items in the PCMC scale by sub-scale domain.**

| | No, Never | Yes, a few times | Yes, most of the time | Yes, all the time | Total *n* |
|---|---|---|---|---|---|
| **Dignity and Respect Subscale** | | | | | |
| 1. Did the doctors and nurses or other staff treat you with respect? | 6 (2%) | 44 (14%) | 100 (33%) | 157 (51%) | 307 |
| 2. Did the doctors, nurses, and other staff at the facility treat you in a friendly manner? | 9 (3%) | 38 (12%) | 107 (35%) | 152 (50%) | 306 |
| 3. Did you feel the doctors, nurses, or other health-care providers shouted at you, scolded, insulted, threatened, or talked to you rudely? | 270 (88%) | 25 (8%) | 8 (3%) | 4 (1%) | 307 |
| 4. Did you feel like you were treated roughly like pushed, beaten, slapped, pinched, physically restrained, or gagged? | 295 (96%) | 9 (3%) | 2 (1%) | 1 (0%) | 307 |
| 5. During examinations in the labor room, were you covered up? | 59 (20%) | 28 (9%) | 65 (22%) | 148 (49%) | 300 |
| 6. Do you feel like your health information was or will be kept confidential at this facility? | 8 (3%) | 41 (14%) | 118 (39%) | 135 (45%) | 302 |
| **Communication and Autonomy Subscale** | | | | | |
| 1. During your time in the health facility did the doctors, nurses, or other health-care providers introduce themselves to you when they first came to see you? | 226 (74%) | 61 (20%) | 17 (6%) | 3 (1%) | 307 |
| 2. Did the doctors, nurses, or other health-care providers call you by your name? | 66 (22%) | 58 (19%) | 70 (23%) | 110 (36%) | 304 |
| 3. Did you feel like the doctors, nurses or other staff at the facility involved you in decisions about your care? | 41 (13%) | 43 (14%) | 109 (36%) | 111 (37%) | 304 |
| 4. During the delivery, do you feel like you were able to be in the position of your choice? | 67 (22%) | 88 (29%) | 53 (18%) | 91 (30%) | 299 |
| 5. Did the doctors, nurses, or other staff at the facility speak to you in a language you could understand? | 1 (0%) | 20 (7%) | 66 (22%) | 219 (72%) | 306 |
| 6. Did the doctors, nurses, or other staff at the facility ask your permission or consent before doing procedures on you? | 47 (15%) | 58 (19%) | 125 (41%) | 74 (24%) | 304 |
| 7. Did the doctors and nurses explain to you why they were doing examinations or procedures on you? | 23 (8%) | 55 (18%) | 148 (48%) | 80 (26%) | 306 |
| 8. Did the doctors and nurses explain to you why they were giving you any medicine? | 33 (11%) | 49 (16%) | 101 (33%) | 121 (40%) | 304 |
| 9. Did you feel you could ask the doctors, nurses, or other staff at the facility any questions you had? | 50 (16%) | 48 (16%) | 124 (41%) | 84 (27%) | 306 |
| **Supportive Care Subscale** | | | | | |
| 1. How did you feel about the amount of time you waited? [†] | 196 (64%) | 42 (14%) | 43 (14%) | 26 (8%) | 307 |
| 2. Did the doctors and nurses at the facility talk to you about how you were feeling? | 35 (11%) | 66 (21%) | 130 (42%) | 76 (25%) | 307 |
| 3. Did the doctors, nurses or other staff at the facility try to understand your anxieties? | 78 (26%) | 74 (25%) | 74 (25%) | 76 (25%) | 302 |
| 4. When you needed help, did you feel the doctors, nurses or other staff at the facility paid attention? | 55 (18%) | 53 (17%) | 117 (38%) | 82 (27%) | 307 |
| 5. Do you feel the doctors or nurses did everything they could to help control your pain? | 70 (23%) | 78 (25%) | 90 (29%) | 69 (22%) | 307 |
| 6. Were you allowed to have someone you wanted (outside of staff at the facility, such as family or friends) to stay with you during labor? | 236 (78%) | 30 (10%) | 25 (8%) | 12 (4%) | 303 |
| 7. Were you allowed to have someone you wanted to stay with you during delivery? | 253 (84%) | 18 (6%) | 25 (8%) | 6 (2%) | 302 |
| 8. Did you feel the doctors, nurses, or other staff at the facility took the best care of you? | 18 (6%) | 48 (16%) | 114 (37%) | 126 (41%) | 306 |
| 9. Did you feel you could completely trust the doctors, nurses, or other staff at the facility with regards to your care? | 14 (5%) | 48 (16%) | 123 (40%) | 122 (40%) | 307 |
| 10. Do you think there were enough health staff in the facility to care for you? | 59 (19%) | 56 (18%) | 113 (37%) | 78 (25%) | 306 |
| 11. Thinking about the labor and postnatal wards, did you feel the health facility was crowded? | 99 (32%) | 48 (16%) | 47 (15%) | 112 (37%) | 306 |
| 12. Thinking about the wards, washrooms, and the general environment of the health facility, would you say the facility was very clean, clean, dirty, or very dirty? [††] | 10 (3%) | 24 (8%) | 127 (41%) | 146 (48%) | 307 |
| 13. Was there water in the facility? | 3 (1%) | 7 (2%) | 26 (8%) | 271 (88%) | 307 |
| 14. Was there electricity in the facility? | 1 (0%) | 0 (0%) | 13 (4%) | 293 (95%) | 307 |
| 15. In general, did you feel safe in the health facility? | 5 (2%) | 16 (5%) | 67 (22%) | 219 (71%) | 307 |

[†]Response options for this question followed the same scale but were: Very short (0), somewhat short (1), somewhat long (2), or long (3).

[††]Response options for this question followed the same scale but were: Very dirty (0), dirty (1), clean (2), very clean (3).

Note: Items for which response rate was lower than the total sample (307) indicate the item was skipped due to refusal, or a "do not know" or "not applicable" response. The procedure for dealing with these is described in the Measures section.

**Table 3. Sub-scale normalized scores.**

| Sub-Scale | Observations | Mean | Standard Deviation | Min | Max |
|---|---|---|---|---|---|
| Dignity and Respect | 307 | 81.65 | 17.64 | 11.11 | 100 |
| Communication and Autonomy | 307 | 58.31 | 18.00 | 7.407 | 100 |
| Supportive care | 307 | 60.31 | 14.96 | 8.888 | 93.33 |

who were literate reported significantly higher levels of PCMC than women who were illiterate or semi-literate ($\beta$ = 5.76, p = 0.006). Women whose delivery was conducted by an unskilled birth attendant reported lower levels of PCMC than women whose delivery was conducted by a Nurse/Midwife/Clinical Officer ($\beta$ = -8.96, p = 0.016). However, the number of observations for this variable is quite small (n = 12), hence need to be interpreted with caution. PCMC was also lower for women with delayed antenatal care, with those having their first antenatal visit in the second ($\beta$ = -5.03, p = 0.006) or third trimester ($\beta$ = -7.29, p = 0.003), reporting lower PCMC scores than women whose first antenatal visit was in the first trimester. Finally, women who were interviewed by phone reported lower PCMC scores ($\beta$ = -7.54, p = 0.006) than those interviewed face-to-face at the health facility. Other variables did not demonstrate significant associations with the PCMC score. In addition, the random intercept suggested meaningful variation in women's PCMC scores across facilities, but facility type (i.e., public, private, faith-based) was not a significant predictor of PCMC scores.

## Discussion

This study sought to assess women's experiences of PCMC and associated factors in a peri-urban setting in Kenya using quantitative methods. We found that PCMC was sub-optimal in this setting. The lowest scores were in the communication and autonomy domain, followed by the supportive care domain—with the highest scores in the dignity and respect sub-domain. The sociodemographic factors associated with PCMC was self-reported literacy, with higher PCMC among literate women compared to illiterate women. Other factors associated with PCMC were timing and frequency of ANC, delivery attendant, and location of interview. The results indicate the need for improvements in PCMC, as well as efforts to address disparities by sociodemographic factors.

The average PCMC score of 58.2 out of 90 found in this study is consistent with scores obtained from other studies in Kenya using the same scale: the scores from the rural and urban county used in the validation of the scale were 59.5 and 60.2 respectively [24]. It is encouraging that most women within the peri-urban setting reported being treated with respect most or all the time (84%; See Table 2). Additionally, the proportion of women reporting physical abuse (5%) and verbal abuse (10%) was low when compared to earlier studies that had estimated physical abuse at 20% [3]. This could be due to the implementation of interventions to improve respectful maternity care in Kenya in the last few years [30]. More work is however needed—especially in the other domains of PCMC.

The low scores in the communication and autonomy domain is also consistent with prior studies on PCMC conducted in Kenya. For example, most respondents (74%) in this study reported that the health care workers never introduced themselves, and about one fifth (22%) reported that the health care workers did not call them by name. This is similar to the findings from the rural and urban Kenya study, where 77% and 85% of respondents respectively reported that providers never introduced themselves; and 27% and 44% respectively reported providers never called them by name [15] Women in our sample were, however, more likely to be able to give birth in the position of their choice: 24% of women in our sample reported

**Table 4. Multilevel model examining associations between PCMC score and selected factors.**

| VARIABLES | Mean PCMC Score | Confidence Interval | p value of coefficient |
|---|---|---|---|
| Age | 0.025 | [-0.304–0.355] | 0.881 |
| Reads very well | 5.758 | [1.671–9.846] | 0.006 |
| *Education of Partner (ref = none)* | | | |
| Post-primary/Vocational/College | -3.446 | [-7.258–0.367] | 0.076 |
| College or above | -4.526–4.402 | [-9.354–0.302] | 0.066 |
| No partner | -0.444 | [-5.645–4.758] | 0.867 |
| *Urban Wealth Quintile (ref = poor or poorest)* | | | |
| Middle | 3.802 | | 0.034 |
| Richer or Richest | | | 0.157 |
| Employed | 0.035 | [-4.404–4.474] | 0.988 |
| Employed in a Health Facility | -3010 | [-7.805–1.785] | 0.219 |
| Experienced Domestic Violence | -1.381 | [-4.279–1.517] | 0.350 |
| High Empowerment | 4.018 | [-0.916–8.951] | 0.110 |
| Experienced Severe Complications | -3.386 | [-7.365–0.593] | 0.095 |
| Delivered in hospital for previous birth | 0.745 | [-2.509–3.999] | 0.654 |
| *Highest Skilled Delivery Provider present at delivery (ref = Nurse/Midwife/Clinical Officer)* | | | |
| Unskilled person or auxiliary nurse/midwife | -8.962 | [-16.247–-1.677] | 0.016 |
| Doctor | 2.645 | [-0.336–5.626] | 0.082 |
| *Trimester of first Antenatal visit (ref = first)* | | | |
| Second | -5.03 | [-8.626–-1.434] | 0.006 |
| Third | -7.288 | [-12.029–-2.546] | 0.003 |
| 4 or more Antenatal visits | 0.340 | [-2.762–3.442] | 0.830 |
| 5 weeks or more of postpartum | -2.708 | [-6.032–0.615] | 0.110 |
| *Religion (ref = Catholic)* | | | |
| Protestant/Pentecostal | 0.620 | [-3.901–5.142] | 0.788 |
| Other Christian | -0.106 | [-3.579–3.367] | 0.952 |
| Muslim, other or refused | 4.592 | [-4.525–13.709] | 0.324 |
| *Tribe (ref = Luo)* | | | |
| Kikuyu | -0.737 | [-4.996–3.523] | 0.735 |
| Luhya | -0.796 | [-4.955–3.362] | 0.707 |
| Kamba | -2.575 | [-7.171–2.021] | 0.272 |
| Other, refused | -3.118 | [-7.843–1.607] | 0.196 |
| *Interview Location (ref = Health Facility)* | | | |
| Community/Home | -4.857 | [-10.697–0.984] | 0.103 |
| Phone | -7.535 | [-12.891–-2.180] | 0.006 |
| *Health Facility(ref = Public)* | | | |
| Faith-based | 13.678 | [-0.137–27.494] | 0.052 |
| Private | 7.153 | [-3.821–18.127] | 0.201 |
| Variance of Random Effects (Health Facility) | 21.87 | [3.40–140.609] | |
| Variance of Residuals | 121.474 | [102.69–143.69] | |
| Observations | 307 | | |
| N of groups (health facilities) | 8 | | |
| Intraclass Correlation (health facility) | 0.176 | [.055–.439] | |

that they were never able to be in the birthing position of their choice, compared to 70% and 39% respectively, for women in the rural and urban Kenyan samples [16]. Reasons provided for poor communication in prior studies include the work environment of providers not

allowing sufficient time to communicate, provider knowledge and assumptions, as well as women's inability to demand or command effective communication and respect for their autonomy [31].

Supportive care was also sub-optimal. In particular, a large proportion of respondents were not allowed to have a companion during labor (78%) and delivery (84%). This is worse than that from other studies assessing companionship during labor and delivery. For example, Afulani, et al (2018) found that 32% of women were not allowed continuous support during labor (with 19% never allowed a companion during labor), while 70% were not allowed continuous support during delivery (61% were never allowed a companion during delivery) [15]. This study also found that although providers often denied women companions at the time of delivery, this was consistent with the preference of some women, who desired support during labor and after delivery, but not during the delivery [15]. Women's reasons for not desiring a companion during their delivery included lack of privacy and not wanting to be seen in their most vulnerable moment by non-health care workers [15]. Further contributing to the sub-optimal supportive care is poor pain control, with about one third of respondents reporting that the health care workers did not do everything they could to control their pain.

Literacy, a proxy indicator for education, has been established as consistently and strongly associated with delivery in a health facility [24,32]. In the current study, literacy was also significantly associated with PCMC. Women who are literate may be more able to communicate effectively with health care workers and negotiate for better experiences at the health facility. Literate women are also more likely to be familiar with the health care provision infrastructure and able to navigate it better [32]. Further, health care workers may provide better treatment to literate women because they are more likely to be able to hold them accountable [31]. Receipt of higher PCMC by more literate women may therefore contribute to the higher facility deliveries among them. Women's education is also associated with improved health-seeking behavior through health awareness, economic autonomy and the ability to make appropriate health decisions [33,34].

Timing and frequency of ANC reflect the level of engagement with the health care system. The later a woman's first visit, the more negative her PCMC scores were, suggesting more negative experiences during childbirth. Other studies conducted in Kenya have similarly shown that ANC timing is associated with experience of care, with women who received ANC in the third trimester reporting poorer experiences [35]. Early ANC visits is associated with positive maternal health outcomes primarily because women receive timely care for preventing or identifying and managing complications [36]. Women who seek ANC early are also able to establish a relationship with their healthcare providers, which can contribute to a better experience during childbirth. Studies in Kenya and other LMIC settings have, however, demonstrated key gaps in quality of ANC, with low SES women having poorer experiences during ANC, which could affect their decision on where to give birth [35].

Births being assisted by non-skilled attendants in a health facility demonstrates a failure in the health system. Although this represented less than 4% of our respondents, other studies have suggested unskilled providers sometimes play clinical roles in facilities including assisting with births. This is sometimes due to overcrowding in maternity units and long wait times due to shortage of clinical staff [12]. Furthermore, that women who were assisted by unskilled providers received poorer PCMC may indicate poorer knowledge of PCMC among this cadre of staff leading to poor PCMC provision [31]. Prior studies have shown that while these unskilled providers may sometimes serve as advocates for women, they can also be perpetrators of abuse [37]. However, given the very small proportion of women who were assisted by unskilled providers in our sample, this finding should be interpreted with caution and explored further in future studies.

The findings based on location of interview seems to demonstrate that women who were on their "home turf" so to speak—a location where they feel more comfortable or empowered (home and community)—were potentially more honest or forthcoming about their negative experiences in the health care setting, when compared with women who were interviewed at the health facility. Women interviewed in the facility may be hesitant to speak negatively about their experience in the facility for fear of retribution by health care workers. This is consistent with other studies where women interviewed at home reported higher PCMC than those interviewed in health facilities, although interviews were face to face in both locations [6,16].

## Strengths and limitations

A key limitation of this study is that the data are from surveys where women self-report their experiences. Thus, like all self-reported data, the findings are prone to social desirability and recall bias. Women may have been reluctant to report negative experiences of care or may not remember all their experiences during childbirth. The strength of this study was that we used a validated tool that has been used in several low- and middle-income contexts and applied it to an under researched context such as peri-urban contexts in cities. Another potential limitation is the mix of both in-person and phone interviews due to the COVID-19 crisis. Although not initially planned, this has helped provide evidence on the feasibility of using the PCMC scale in phone interviews. Finally, the unique sample implies the findings may not be generalizable to other settings in Kenya.

## Conclusions

This study is among the few in low-resource settings on women's perspectives on person-centered maternity care within a peri-urban setting. Our findings support evidence of poor PCMC across all domains but particularly with regards to communication and autonomy and supportive care. These results indicate the need for interventions to improve in PCMC at health facilities in peri-urban settings in Kenya. Given the unique context of peri-urban settings, it is important to examine interventions that are feasible and relevant in these settings. This will help ensure peri-urban women are not left behind in efforts to improve maternal and neonatal outcomes.

## Supporting information

**S1 Appendix. Participating hospitals.**
(DOCX)

**S2 Appendix. Questionnaire on person centered maternity care.**
(XLSX)

## Acknowledgments

We would like to thank Christine Achieng, Edwina Ndhine, Florence Okeyo King and Joy Minyenya-Njuguna who participated in collecting the data for the study. We are grateful to the Ford program and the Kellogg Institute for International studies for supporting the study. We thank the leadership of the health department at Nairobi County for giving us permission to recruit respondents from the public health facilities under their supervision and the different counties within Embakasi where we collected our data. Most importantly we thank the women of Embakasi for sharing their birth stories with us and allowing us to conduct this study.

## Author Contributions

**Conceptualization:** Jackline Oluoch-Aridi, Patience Afulani, Cindy Makanga, Danice Guzman, Laura Miller-Graff.

**Data curation:** Cindy Makanga, Danice Guzman.

**Formal analysis:** Jackline Oluoch-Aridi, Patience Afulani, Danice Guzman.

**Funding acquisition:** Jackline Oluoch-Aridi, Danice Guzman, Laura Miller-Graff.

**Investigation:** Jackline Oluoch-Aridi, Cindy Makanga, Danice Guzman, Laura Miller-Graff.

**Methodology:** Jackline Oluoch-Aridi, Patience Afulani, Cindy Makanga, Danice Guzman, Laura Miller-Graff.

**Project administration:** Jackline Oluoch-Aridi, Cindy Makanga, Danice Guzman, Laura Miller-Graff.

**Resources:** Laura Miller-Graff.

**Software:** Danice Guzman.

**Supervision:** Jackline Oluoch-Aridi, Cindy Makanga.

**Validation:** Jackline Oluoch-Aridi, Patience Afulani, Danice Guzman, Laura Miller-Graff.

**Writing – original draft:** Jackline Oluoch-Aridi.

**Writing – review & editing:** Patience Afulani, Danice Guzman, Laura Miller-Graff.

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
