## [Decision Letter · Decision Letter 0]

4 Feb 2021

PONE-D-21-00183

Examining Person-Centered Maternity Care in peri-urban settings in Nairobi, Kenya.

PLOS ONE

Dear authors,

I hope your are well. You paper has been reviewed by two reviewers and they have raised a number of important points that you can address. Beside these, I have a number of points to report on your manuscript.First and foremost, your manuscript is now very difficult to follow, mostly because of poor quality of writing. I would consider your paper for publication if you could revise your paper through a professional proof reader and a substantial improvement has been made on the writing quality before resubmission. I am also concerned with the study methodology. At the first line of the methodology section, you mentioned that your study is part of a large scale longitudinal data though later you mentioned three months time when you conducted your study. How could it possible to conduct a longitudinal study in three months? It was also difficult for me to follow other sections of the methodology. I would recommend you for another comprehensive look to make them readable and easily understandable. I will do another comprehensive review if all points have been addressed and resubmitted.

We look forward to receiving your revised manuscript.

Kind regards,

Md Nuruzzaman Khan

Academic Editor

PLOS ONE

Journal Requirements:

2. In the Methods section of the manuscript please include additional information on the following:

1) please provide a justification for the sample size used in your study, including any relevant power calculations (if applicable).

2)  Please include in your Methods section (or in Supplementary Information files) the participating hospitals/institution.

5. We noticed you have some minor occurrence of overlapping text with the following previous publication(s), which needs to be addressed:

- https://reproductive-health-journal.biomedcentral.com/articles/10.1186/s12978-017-0449-4

- https://journals.plos.org/plosmedicine/article?id=10.1371%2Fjournal.pmed.1001847

The text that needs to be addressed primarily involves the Introduction. In your revision ensure you cite all your sources (including your own works), and quote or rephrase any duplicated text outside the methods section. Further consideration is dependent on these concerns being addressed.

Reviewers' comments:

Reviewer's Responses to Questions

**Comments to the Author**

1. Is the manuscript technically sound, and do the data support the conclusions?

Reviewer #1: Yes

Reviewer #2: Yes

2. Has the statistical analysis been performed appropriately and rigorously? 

Reviewer #1: Yes

Reviewer #2: Yes

3. Have the authors made all data underlying the findings in their manuscript fully available?

Reviewer #1: Yes

Reviewer #2: Yes

4. Is the manuscript presented in an intelligible fashion and written in standard English?

Reviewer #1: No

Reviewer #2: Yes

5. Review Comments to the Author

Reviewer #1: The manuscript is well formatted and represent the person-centered maternity healthcare in peri-urban area of Kenya. The concept is quite interesting and up to date with the recent health care services. The authors describe the use of pre-developed scales, sampling and result discussion which is very effective and ease to understand. Overall, the work is great except some following problems-

1. Peri-urban area is mainly worked as the bridge of rural and urban area. So, basically there must be some difference in the maternal health-facilities. In the introduction part, if the author(s) can add a brief rationale of why he(they) uses the peri-urban area for person centered maternity healthcare, the study might be justified by the study location.

2. Several part of the manuscript can be modified to represent the concise information, specially, in the introduction and statistical analysis of methodology part. Note that, several repetitive words are seen in the manuscript, in the consequences, several parts are poorly matched with the total manuscripts (for example, please see the 3rd and 4th paragraph of the introduction section and full methods section).

3. The author can describe the person-centered maternity care. Although he defined it with the dimensions that constructed from two papers, but a brief explanation of such dimensions could be included.

4. In table 1, the number of never married is 45 while in education level of partner implies that there are 41 counts have no partner. In this case, the author should explain the bit about the variation. Again, what is pregnancy complication and severe pregnancy complications? The author should clarify the pregnancy complication and severe pregnancy complication and how he used these two attributes? Otherwise, the author should delete one of the variables or clarify the use of similar variables. Lastly, in the variable, type of facility, the author(s) mentioned the mission is one of the types while in the methods part he mentions the faith-based facilities. In this regard, the author(s) should be more consistent with the variables and the labels.

5. Table 2 can be regenerated by increasing the columns instead of rows. Additionally, table 3 can be integrated with the table 2.

6. The conclusion should be more result oriented.

7. Grammatical and repetitive words should be declined to the concise information related to the study objectives and results.

Reviewer #2: Comments on Title

This study is completed in one area of Nairobi County, so how can it represent the whole Nairobi County as well as the Kenya? That means how do you measure the whole County and/or Country by only one area/city?

You may rephrase your title as: ‘Examining Person-Centered Maternity Care in peri-urban settings in Embakasi, Nairobi, Kenya’

Comments on Abstract

i. In introduction section you have wrote ‘like’ before the word ‘Kenya’. I recommend you to recheck that word and sentence. I think ‘in’ will replace the ‘like’.

ii. In methods section you have considered the women aged 18-49 years who had delivered a baby within 4 to 6 weeks as the respondents. But you didn’t provide the any rationale behind these. Why did you choose 18-49 years, and within 4 to 6 weeks?

iii. What was your logic behind recruiting the women from public (n=118), private (n=76), and faith based (n=113) health facilities?

Comments on Introduction Section

a. Authors wrote pointlessly a lot about background profile of the study. Please make it short, simple and precise according to the title.

b. Add some statistics nationally and inter nationally, and compare them.

c. Need a major revision in the Literature review section.

d. Draw a clear research gap.

Comments on Methods Section

1. Is this study a longitudinal or cross-sectional study? In abstract section you wrote it is cross-sectional but in methods section you have mentioned it is longitudinal. So this is confusing. Make it clear.

2. Add the reference(s) after the following sentences:

The area is characterized by low-income ……. access to water and waste disposal.

The health system within Embakasi ……health facilities and faith-based health facilities.

The main referral health facility for …… that is situated in Embakasi-West.

We divided the Embakasi area into its ……… types of health facilities in the setting.

But we decided to retain them because …….. with the prior studies conducted in Kenya.

3. Simply clear the rationale of using the multistage purposive sampling approach, and simple Ordinary Least Squares (OLS) regression.

4. Give a clear idea about the Dependent Variable: The person-centered maternity score (PCMC Score).

5. How did you test the collinearity problem? Please explain.

Comments on Results Section

a. Authors need to re-category the following study variables in table 1: Education Level, Education Level of Partner, Literacy, Partner's Occupation, Highest skilled person present during delivery. In these variables the sub-categories are overlapping. As for example in Education Level authors have used three sub-categories named ‘No School/Primary’, ‘Post-primary/Vocational/College’, and ‘College or above’. In this case how did the authors consider the No School and Primary in one category? Also the 2nd and 3rd options include College, so these are more puzzling.

b. What is the difference between Pregnancy Complications and severe Pregnancy Complications? You may point out some name of the Pregnancy Complications and severe Pregnancy Complications.

c. In table 2 and 3 some variables don’t represent the exact total figure (307). Why? Make it clear.

d. In table 4 authors used REML. What does it mean? Add the meaning of ‘REML’ in notes section under the table 4 and in abbreviation list.

e. In the note section under the table 4 authors used ‘*** p<0.001’, this is unnecessary.

f. Authors may add an extra column in table 1 & 4 to indicate the p-values.

Comments on Discussion Section

i. Authors may write Embakasi instead of Kenya in 2nd line of the 1st paragraph of the Discussion Section.

ii. Add the reference(s) after the following sentences:

It is encouraging to note that …… that health care workers treated them with respect.

The prior studies in rural and urban Kenya ….. providers never called them by name.

Also related to the sub-optimal supportive ……. they could to control their pain.

iii. What do LMIC and SES mean? Clear the meaning of them and add in abbreviation list.

Final Comments

The results indicated that women who were literate reported significantly higher levels of PCMC than women who were illiterate or semi-literate. Women whose delivery was undertaken by an unskilled birth attendant reported lower levels of PCMC than women whose delivery was conducted by a Nurse/Midwife/Clinical Officer. PCMC was also lower for women with delayed antenatal care, with those having their antenatal clinic visit in the second or third trimester, reporting lower PCMC scores than women whose first antenatal visit was in the first trimester. Finally, women who were interviewed by phone reported PCMC scores that were lower than those interviewed face-to-face at the health facility.

It is pretty obvious that literate women whose delivery was undertaken by a Nurse/Midwife/Clinical Officer and whose first antenatal visit was in the first trimester will show high PCMC scores than the others. This is a well-known fact. So, why the authors have tried to justify these findings?

6. PLOS authors have the option to publish the peer review history of their article (what does this mean?). If published, this will include your full peer review and any attached files.

Reviewer #1: **Yes: **Billah, Md. Arif

Reviewer #2: **Yes: **Md. Shariful Islam, Lecturer, Department of Public Health, First Capital University of Bangladesh, Chuadanga, Khulna, Bangladesh

---

## [Author Response · Author response to Decision Letter 0]

16 Mar 2021

RESPONSES TO REVIEWER #1

 COMMENTS RESPONSES Section in the Manuscript

1. Peri-urban area is mainly worked as the bridge of rural and urban area. So, basically there must be some difference in the maternal health-facilities. In the introduction part, if We can add a brief rationale of why he (they) uses the peri-urban area for person centered maternity healthcare, the study might be justified by the study location. We concur with the reviewer that there are some differences in terms of maternal health facilities serving peri-urban settings and urban/rural areas. We have included a brief rationale in the introduction to clarify on why we focused our study within a peri-urban setting. This is mainly related to differences in quality of maternal health services. See pasted below: 

“...Although previous studies have examined PCMC in both rural and urban settings in Kenya, little work has been done in peri-urban settings. Peri-urban settings in Kenya are often close to cities, but lack access to basic amenities such as running water and adequate sanitation. The healthcare system within these settings has also been reported to be deficient and health facilities are known to provide varied quality of care (17)...” Introduction section pg. 3

2. Several part of the manuscript can be modified to represent the concise information, specially, in the introduction and statistical analysis of methodology part. 

Note that, several repetitive words are seen in the manuscript, in the consequences, several parts are poorly matched with the total manuscripts (for example, please see the 3rd and 4th paragraph of the introduction section and full methods section). The manuscript has been revised to represent concise information. The introduction and the methodology have been revised to remove the repetitive words.

The third and fourth paragraph of the introduction section as well as the full methods section have been revised Introduction section pg. 3-5

Methodology section pg. 5-8

3. The author can describe the person-centered maternity care. Although he defined it with the dimensions that constructed from two papers, but a brief explanation of such dimensions could be included. A brief explanation has been included as follows to further describe person centered maternity care : 

Person-centered maternity care (PCMC) represents the experience or interpersonal dimensions of quality of care. It refers to care that is respectful and responsive to the needs of women and their families (5). PCMC emphasizes the patient-provider relationship, and highlights issues such whether women are treated in a dignified and respectful manner, are communicated to effectively and feel involved in decision-making about their care, and feel supported both emotionally and socially during childbirth (5). 

Introduction section pg.3

4. In table 1, the number of never married is 45 while in education level of partner implies that there are 41 counts have no partner. In this case, the author should explain the bit about the variation. 

 We have provided a note with a brief description about the variable marital status and the educational level of the participants in the research in table 1.

6 respondents are cohabitating, meaning they are never married but they have a partner. 2 respondents are widowed, so they do not have a partner but they were married. This variation explains the difference of 4 respondents between these two variables. This has been briefly mentioned in a footnote.

 Pg 8 Foot note 

 Again, what is pregnancy complication and severe pregnancy complications? The author should clarify the pregnancy complication and severe pregnancy complication and how he used these two attributes? Otherwise, the author should delete one of the variables or clarify the use of similar variables. We have provided an explanation distinguishing between pregnancy complications and severe pregnancy complications. There were 23 pregnancy complications that the respondent could have identified, or they had the option to list a complication which was not included in the options of responses. If they listed any complications, they were asked if the problem was severe. So the severity of the problem was self-determined. This has been added in the description of the variables.

 Results section pg. 7

 Lastly, in the variable, type of facility, We mentioned the mission is one of the types while in the methods part he mentions the faith-based facilities. In this regard, Weshould be more consistent with the variables and the labels. The comment is well received and for the variable type of facility, We will use faith-based as the consistent term to describe health facilities that are run and operated by religious organizations Throughout the manuscript

5. Table 2 can be regenerated by increasing the columns instead of rows. Additionally, table 3 can be integrated with the table 2. Table 2 has been regenerated by increasing the columns as requested.

 Unfortunately for table 2 and 3 cannot be integrated because of space Table 2 pg 13-15

6. The conclusion should be more result oriented. The conclusion has been revised to speak to the results Conclusion section pg 20

7. Grammatical and repetitive words should be declined to the concise information related to the study objectives and results. The manuscript has been reviewed and the repetitive words have been removed Throughout the manuscript

RESPONSES TO REVIEWER #2

 COMMENTS RESPONSES 

 TITLE 

1. This study is completed in one area of Nairobi County, so how can it represent the whole Nairobi County as well as the Kenya? That means how do you measure the whole County and/or Country by only one area/city?

You may rephrase your title as: ‘Examining Person-Centered Maternity Care in peri-urban settings in Embakasi, Nairobi, Kenya’ To address this concern while keeping the title concise, we have rephrased the title to: ‘Examining Person-Centered Maternity Care in a peri-urban setting in Nairobi, Kenya. Title page 

 ABSTRACT 

2. In the introduction section you have wrote ‘like’ before the word ‘Kenya’. I recommend you to recheck that word and sentence. I think ‘in’ will replace the ‘like’. We have replaced the word ‘like’ with ‘in’ as advised. Abstract 

3. In the methods section you have considered the women aged 18-49 years who had delivered a baby within 4 to 6 weeks as the respondents. But you didn’t provide any rationale behind these. Why did you choose 18-49 years, and within 4 to 6 weeks? We considered women aged 18-49 years because these are women considered to be of reproductive age.

We considered women who had delivered a baby within 4 to 6 weeks because this is when women go back to hospital for their first clinical visits Pg 5

4. What was your logic behind recruiting the women from public (n=118), private (n=76), and faith based (n=113) health facilities? This sample size was established recruited was based on available health facilities in the setting and a multi stage purposive sampling design strategy that has been discussed in the methodology section 

 INTRODUCTION SECTION 

5. Authors wrote pointlessly a lot about background profile of the study. Please make it short, simple and precise according to the title. The background has been revised to make it short and simple Introduction section 

6. Add some statistics nationally and internationally, and compare them. Authors have added a comparison to the maternal mortality rate of low income countries that was estimated as 462 per 100, 000 live births and compared it to Kenya’s national mortality rate of 362 per 100,000 to provide a comparison First paragraph pg 1 

7. Need a major revision in the Literature review section. The literature review section has been revised Introduction section 

8. Draw a clear research gap. The research gap has been identified as a lack of information on the extent of PCMC in peri-urban settings that are contributing to maternal health deaths Introduction section pg 1 last paragraph

 METHODS SECTION 

9. Is this study a longitudinal or cross-sectional study? In abstract section you wrote it is cross-sectional but in methods section you have mentioned it is longitudinal. So this is confusing. Make it clear. This has been corrected, the study was a cross sectional study Page 5

10. Add the reference(s) after the following sentences:

The area is characterized by low-income ……. access to water and waste disposal.

The health system within Embakasi ……health facilities and faith-based health facilities.

The main referral health facility for …… that is situated in Embakasi-West.

We divided the Embakasi area into its ……… types of health facilities in the setting.

But we decided to retain them because …….. with the prior studies conducted in Kenya. Citations have been included in the relevant areas 

11. Simply clear the rationale of using the multistage purposive sampling approach, and simple Ordinary Least Squares (OLS) regression. The sampling strategy that we used was multistage purposive sampling approach rather than purposive. We took samples in stages using smaller and smaller sampling units at each stage. We then randomly sampled the women in each stage at the selected health facilities within the sub-County.

We used LS regression because the dependent variable was a continuous variable with a normal distribution. Page 5 

12. Give a clear idea about the Dependent Variable: The person-centered maternity score (PCMC Score). More detail has been provided about the dependent variable the PCMC score as seen pasted below 

The PCMC scale is a validated 30-item scale with three sub-scales i) dignity and respect, ii) autonomy and communication and iii) supportive care. Each item is on a 4-point response scale – 0 “no, never”, 1: “yes, a few times”, 2: “yes, most of the time”, and 3: “yes all the time.” The full list of items are provided in (additional file 1). ……

Summing response to items (after reverse coding negatively worded items) yields a score range of 0 to 90, with a lower score implying poorer PCMC. To account for missing responses to certain questions (which were not applicable to certain women),- the scores were calculated using a running mean across items, and then converted to the typical summed score (0 to 90) to enable comparisons to previously published work on the scale (25),(28). We used a similar approach to generate sub-scale scores. All sub-scale scores were standardized to range from 0 to 100 to enable comparisons across sub-scales.

 Page 6 and 7

13. How did you test the collinearity problem? Please explain. Collinearity was tested by calculating the centered variance inflation factor (VIF) as a post-estimation. We applied the rule of thumb from Chatterjee and Hadi (2012), meaning that there was evidence of collinearity if, of all the coefficients included in the model, the largest VIF is greater than 10. In our final model, the largest VIF is 3.85. We have added a reference to the VIF in the text and the acronym list. Page 8 paragraph 1 & Acronym list 

 RESULTS SECTION 

14. Authors need to re-category the following study variables in table 1: Education Level, Education Level of Partner, Literacy, Partner's Occupation, Highest skilled person present during delivery. In these variables the sub-categories are overlapping. As for example in Education Level authors have used three sub-categories named ‘No School/Primary’, ‘Post-primary/Vocational/College’, and ‘College or above’. In this case how did the authors consider the No School and Primary in one category? Also the 2nd and 3rd options include College, so these are more puzzling. The issue with education (of self and partner) was an error in labelling. The 2nd option is defined as post-primary, vocational, and secondary. The 3rd option is college or university. We thank the reviewers for identifying this; it has been corrected in Table 1. No school and primary was considered as one category due to low frequencies of response to “no school” option (only 2 respondents and 1 partner). Partner’s education responses were re-classified categories when response rates to specific options such “agricultural worker” or “unemployed” were particularly low. In the survey, the subcategories are mutually exclusive and do not overlap. For highest skilled person present at delivery, the categories are 1) Low or no skill (Auxiliary nurse, auxiliary midwife, or no skilled person); 2) Some skill (Clinical officer, nurse, midwife); 3) Very high skill (Doctor). We understand that the labels were confusing so they have been corrected to show that the subcategories are not overlapping. Table 1

15. What is the difference between Pregnancy Complications and severe Pregnancy Complications? You may point out some name of the Pregnancy Complications and severe Pregnancy Complications. We have provided an explanation distinguishing between pregnancy complications and severe pregnancy complications. There were 23 pregnancy complications that the respondent could have identified, or they had the option to list a complication which was not included in the options of responses. If they listed any complications, they were asked if the problem was severe. So the severity of the problem was self-determined. This has been added in the description of the variables.

 Results Section - Page 7

16. In table 2 and 3 some variables don’t represent the exact total figure (307). Why? Make it clear. In Table 2, some variables do not represent the exact total figure due to missing responses to individual questions, or questions which were not applicable to certain respondents (for example, if a woman had a scheduled caesarian birth, the question of choice of delivery position is not relevant to her). This is described in the Measures section, PCMC subsection (“To account for missing responses…”). We have included a note below the table which repeats this information. Note below Table 2

17. In table 4 authors used REML. What does it mean? Add the meaning of ‘REML’ in notes section under the table 4 and in abbreviation list. REML means the model was fit via restricted maximum likelihood. This was already referenced in the text but the acronym has been added to the text and abbreviation list. It is not needed in Table 4 so it has been removed. Abbreviation list 

18. In the note section under the table 4 authors used ‘*** p<0.001’, this is unnecessary. A column for p value has been added and the asterisks have been removed accordingly. Tables

19. Authors may add an extra column in table 1 & 4 to indicate the p-values. A column for p value has been added and the asterisks have been removed accordingly. Table 1 and 4 

 DISCUSSION SECTION 

20. Authors may write Embakasi instead of Kenya in 2nd line of the 1st paragraph of the Discussion Section. 

21. Add the reference(s) after the following sentences:

It is encouraging to note that …… that health care workers treated them with respect.

The prior studies in rural and urban Kenya ….. providers never called them by name.

Also related to the sub-optimal supportive ……. They could to control their pain. The sentence “…It is encouraging to note that …… that health care workers treated them with respect….” Represents the results from the current study and hence does not need to be cited

The relevant citation (32) for this sentence has been included in the text

The prior studies in rural and urban Kenya ….. providers never called them by name.

A citation has been included for the phrase “…Also related to the sub-optimal supportive ……. They could to control their pain….’’ Pg 17

Pg 18 

Pg 18

22. What do LMIC and SES mean? Clear the meaning of them and add in abbreviation list. We have included this in the abbreviation list.

LMIC- low and middle-income countries

SES- Socioeconomic status Abbreviation list 

 FINAL COMMENTS 

23. The results indicated that women who were literate reported significantly higher levels of PCMC than women who were illiterate or semi-literate. Women whose delivery was undertaken by an unskilled birth attendant reported lower levels of PCMC than women whose delivery was conducted by a Nurse/Midwife/Clinical Officer. PCMC was also lower for women with delayed antenatal care, with those having their antenatal clinic visit in the second or third trimester, reporting lower PCMC scores than women whose first antenatal visit was in the first trimester. Finally, women who were interviewed by phone reported PCMC scores that were lower than those interviewed face-to-face at the health facility.

It is pretty obvious that literate women whose delivery was undertaken by a Nurse/Midwife/Clinical Officer and whose first antenatal visit was in the first trimester will show high PCMC scores than the others. This is a well-known fact. So, why the authors have tried to justify these findings? We noted this to highlight that our findings are consistent with previous evidence that emphasizes the role of literacy as a factor related to high PCMC in a peri-urban setting 

JOURNAL REQUIREMENTS

 COMMENTS RESPONSES PLACE IN MANUSCRIPT

1. Please ensure that your manuscript meets PLOS ONE's style requirements, including those for file naming. We have ensured that the manuscript meets the PLOS One style requirements. The file names have been adjusted accordingly Throughout the manuscript

2. In the Methods section of the manuscript please include additional information on the following:

1) please provide a justification for the sample size used in your study, including any relevant power calculations (if applicable).

2) Please include in your Methods section (or in Supplementary Information files) the participating hospitals/institution. We have included in an additional file the participating hospitals.

1)The sample was based on the available health facilities in the Embakasi area

2)The facilities were selected to represent the three categories of faith-based health facilites-Ruben Centre Clinic, public health facilities-Mama Lucy Kibaki Hospital and private health facilities-Pipeline Nursing Home, Dandora Medical Centre, Samaritan Medical Services, Mkunga Hospital, Provide International Clinic, Paradise Health Clinic. This information has been included in a supplementary information file. Methods section

We have changed this and have put in a data statement that responds to the data repository where we have shared deidentified data 

4. Your ethics statement should only appear in the Methods section of your manuscript. If your ethics statement is written in any section besides the Methods, please move it to the Methods section and delete it from any other section. Please ensure that your ethics statement is included in your manuscript, as the ethics statement entered into the online submission form will not be published alongside your manuscript. We have included the ethics statement in the Methods section of the manuscript.

Ethics approval for the study was provided by the Strathmore University Institutional Ethics Review Committee (SU-IERC), the National Commission for Science and Technology (NACOSTI) as well as the Country Directors of health in charge of the sub-county.

 Methods section

5. We noticed you have some minor occurrence of overlapping text with the following previous publication(s), which needs to be addressed:

- https://reproductive-health-journal.biomedcentral.com/articles/10.1186/s12978-017-0449-4

- https://journals.plos.org/plosmedicine/article?id=10.1371%2Fjournal.pmed.1001847

The text that needs to be addressed primarily involves the Introduction. In your revision ensure you cite all your sources (including your own works), and quote or rephrase any duplicated text outside the methods section. Further consideration is dependent on these concerns being addressed. The overlapping text that is mentioned in the reproductive health journal are standard established frameworks of describing disrespect and abuse. We have re-framed and re-phrased the introduction with the overlapping added in text explaining why we chose to highlight the Bowser and Hill (2010) frameworks that establish disrespect and abuse. Introduction section

---

## [Decision Letter · Decision Letter 1]

13 Apr 2021

PONE-D-21-00183R1

Examining Person-Centered Maternity Care in a peri-urban setting in Kenya.

PLOS ONE

Dear Dr. Oluoch-Aridi,

Thank you for submitting your manuscript to PLOS ONE. After careful consideration, we feel that it has merit but does not fully meet PLOS ONE’s publication criteria as it currently stands. Therefore, we invite you to submit a revised version of the manuscript that addresses the points raised during the review process.

We look forward to receiving your revised manuscript.

Kind regards,

Md Nuruzzaman Khan

Academic Editor

PLOS ONE

Journal Requirements:

Please review your reference list to ensure that it is complete and correct. If you have cited papers that have been retracted, please include the rationale for doing so in the manuscript text, or remove these references and replace them with relevant current references. Any changes to the reference list should be mentioned in the rebuttal letter that accompanies your revised manuscript. If you need to cite a retracted article, indicate the article’s retracted status in the References list and also include a citation and full reference for the retraction notice

Reviewers' comments:

Reviewer's Responses to Questions

**Comments to the Author**

1. If the authors have adequately addressed your comments raised in a previous round of review and you feel that this manuscript is now acceptable for publication, you may indicate that here to bypass the “Comments to the Author” section, enter your conflict of interest statement in the “Confidential to Editor” section, and submit your "Accept" recommendation.

Reviewer #1: All comments have been addressed

Reviewer #2: (No Response)

2. Is the manuscript technically sound, and do the data support the conclusions?

Reviewer #1: Yes

Reviewer #2: Yes

3. Has the statistical analysis been performed appropriately and rigorously? 

Reviewer #1: Yes

Reviewer #2: Yes

4. Have the authors made all data underlying the findings in their manuscript fully available?

Reviewer #1: Yes

Reviewer #2: Yes

5. Is the manuscript presented in an intelligible fashion and written in standard English?

Reviewer #1: Yes

Reviewer #2: Yes

6. Review Comments to the Author

Reviewer #1: Authors of this manuscript explained all the answers in their latest version. This could be publishable. Additionally author(s) should be careful on results representations.

Reviewer #2: Comments on Title

This study is completed in one area of Nairobi County, so how can it represent the whole Nairobi County as well as the Kenya? That means how do you measure the whole County and/or Country by only one area/city? I think the following title would be the suitable for this paper: ‘Examining Person-Centered Maternity Care in peri-urban settings in Embakasi, Nairobi, Kenya’.

Comments on Abstract

i.In methods section you have considered the women aged 18-49 years who had delivered a baby within 4 to 6 weeks as the respondents. I have asked you that: Why did you choose 18-49 years? You have told it is a reproductive age. But World Health Organization has said that the reproductive age for a female is 15-49 years. So what’s your argument?

ii.What was your logic behind recruiting the women from public (n=118), private (n=76), and faith based (n=113) health facilities? This is still unexplained. There is nothing in methods section about the selection of women from public (n=118), private (n=76), and faith based (n=113).

Comments on Methods Section

1.The authors didn’t provide the reference(s) after the following sentences.

The area is characterized by low-income ……. access to water and waste disposal.

The health system within Embakasi ……health facilities and faith-based health facilities.

We divided the Embakasi area into its ……… types of health facilities in the setting.

2.In the text write the acceptance/rejection range of variance inflation factor (VIF) in case of collinearity problem.

Comments on Discussion Section

i.Authors may write Embakasi instead of Kenya in 2nd line of the 1st paragraph of the Discussion Section.

ii.The authors didn’t provide the reference(s) after the following sentences.

Also related to the sub-optimal supportive ……. they could to control their pain.

7. PLOS authors have the option to publish the peer review history of their article (what does this mean?). If published, this will include your full peer review and any attached files.

Reviewer #1: **Yes: **Md Arif Billah

Reviewer #2: **Yes: **Md. Shariful Islam, Lecturer, Department of Public Health, First Capital University of Bangladesh, Chuadanga, Khulna, Bangladesh

---

## [Author Response · Author response to Decision Letter 1]

4 Jun 2021

RESPONSE TO REVIEWERS

REVIEWER #1

Comment Response

Authors of this manuscript explained all the answers in their latest version. This could be publishable. Additionally author(s) should be careful on results representations. This is well appreciated. The results have been revised for clarity

REVIEWER #2

Comment Response

Comments on Title 

This study is completed in one area of Nairobi County, so how can it represent the whole Nairobi County as well as the Kenya? That means how do you measure the whole County and/or Country by only one area/city? I think the following title would be the suitable for this paper: ‘Examining Person-Centered Maternity Care in peri-urban settings in Embakasi, Nairobi, Kenya’. The title has been changed to ‘Examining Person-Centered Maternity Care in peri-urban settings in Embakasi, Nairobi, Kenya’.

Comments on Abstract 

In methods section you have considered the women aged 18-49 years who had delivered a baby within 4 to 6 weeks as the respondents. I have asked you that: Why did you choose 18-49 years? You have told it is a reproductive age. But World Health Organization has said that the reproductive age for a female is 15-49 years. So what’s your argument? We focused on women between the age of 18 to 49 because the age 18 years is the legal age of obtaining consent. In Kenya 15 year olds are considered children and would have required us to seek out their parents and this would have been difficult.

What was your logic behind recruiting the women from public (n=118), private (n=76), and faith based (n=113) health facilities? This is still unexplained. There is nothing in methods section about the selection of women from public (n=118), private (n=76), and faith based (n=113). This was based on purposive sampling ( See page 8). We have edited this in the body of the manuscript to indicate that this was to reflect women’s experiences across all types of health facilities present in this area.

Comments on Methods Section 

The authors didn’t provide the reference(s) after the following sentences.

The area is characterized by low-income ……. access to water and waste disposal.

The health system within Embakasi ……health facilities and faith-based health facilities.

We divided the Embakasi area into its ……… types of health facilities in the setting. 

For the quote “… The area is characterized by low-income ……. access to water and waste disposal….” A citation has been included from a UN Habitat publication – The state of African cities (2014) and is number 28 in the reference list.

This statement just provides a description for the different types of health facilities in operation in the Embakasi area 

This is also a description on the administrative divisions of Embakasi 

In the text write the acceptance/rejection range of variance inflation factor (VIF) in case of collinearity problem. This has been provided for in page 10. 

Comments on Discussion Section 

Authors may write Embakasi instead of Kenya in 2nd line of the 1st paragraph of the Discussion Section The authors have changed it to Embakasi

The authors didn’t provide the reference(s) after the following sentences;

Also related to the sub-optimal supportive ……. they could to control their pain. A citation No. 15 has been provided for the statement on supportive care

---

## [Editor Report · Decision Letter 2]

27 Jul 2021

PONE-D-21-00183R2

Examining Person-Centered Maternity Care in a peri-urban setting in Kenya.

PLOS ONE

Dear Dr. Oluoch-Aridi,

I hope you're well. I thank you for your hard work on this manuscript. The current form of this manuscript is acceptable. However, I am not happy with the English, many sentences are very poorly written. Could you please revise your paper one more time for English? You may take this edited from the professional proof reader or get help from any native English speaker. Please include the current version of this manuscript and revised version ( with coloured texts if revision made) when submitting your revised manuscript. 

We look forward to receiving your revised manuscript and read it.

Kind regards,

Dr Md Nuruzzaman Khan

Academic Editor

PLOS ONE

Journal Requirements:

Additional Editor Comments (if provided):

Regards

Dr Md Nuruzzaman Khan

Academic Editor

Plos ONE
---

## [Author Response · Author response to Decision Letter 2]

4 Sep 2021

The editor had required us to do some editorial work. We have reviewed the manuscript thoroughly as per request

---

## [Editor Report · Decision Letter 3]

7 Sep 2021

Examining person-centered maternity care in a peri-urban setting in Embakasi, Nairobi, Kenya

PONE-D-21-00183R3

Dear Dr. Oluoch-Aridi,

We’re pleased to inform you that your manuscript has been judged scientifically suitable for publication and will be formally accepted for publication once it meets all outstanding technical requirements.

Kind regards,

Md Nuruzzaman Khan, PhD

Academic Editor

PLOS ONE
---

## [Editor Report · Acceptance letter]

14 Sep 2021

PONE-D-21-00183R3 

Examining person-centered maternity care in a peri-urban setting in Embakasi, Nairobi, Kenya. 

Dear Dr. Oluoch-Aridi:

I'm pleased to inform you that your manuscript has been deemed suitable for publication in PLOS ONE. Congratulations! Your manuscript is now with our production department. 

Kind regards, 

on behalf of

Dr. Md Nuruzzaman Khan 

Academic Editor

PLOS ONE